# Modified MRI Anonymization (De-Facing) for Improved MEG Coregistration

**DOI:** 10.3390/bioengineering9100591

**Published:** 2022-10-21

**Authors:** Ricardo Bruña, Delshad Vaghari, Andrea Greve, Elisa Cooper, Marius O. Mada, Richard N. Henson

**Affiliations:** 1Center for Cognitive and Computational Neuroscience, Universidad Complutense de Madrid, 28040 Madrid, Spain; 2Department of Radiology, Rehabilitation and Physical Therapy, Universidad Complutense de Madrid, IdISSC, 28040 Madrid, Spain; 3Department of Electrical & Computer Engineering, Tarbiat Modares University, Tehran P.O. Box 14115-111, Iran; 4Medical Research Council Cognition and Brain Sciences Unit, University of Cambridge, Cambridge CB2 7EF, UK; 5Department of Psychiatry, University of Cambridge, Cambridge CB2 OSZ, UK

**Keywords:** deidentification, data sharing, magnetic resonance imaging, magnetoencephalography

## Abstract

Localising the sources of MEG/EEG signals often requires a structural MRI to create a head model, while ensuring reproducible scientific results requires sharing data and code. However, sharing structural MRI data often requires the face go be hidden to help protect the identity of the individuals concerned. While automated de-facing methods exist, they tend to remove the whole face, which can impair methods for coregistering the MRI data with the EEG/MEG data. We show that a new, automated de-facing method that retains the nose maintains good MRI-MEG/EEG coregistration. Importantly, behavioural data show that this “face-trimming” method does not increase levels of identification relative to a standard de-facing approach and has less effect on the automated segmentation and surface extraction sometimes used to create head models for MEG/EEG localisation. We suggest that this trimming approach could be employed for future sharing of structural MRI data, at least for those to be used in forward modelling (source reconstruction) of EEG/MEG data.

## 1. Introduction

There is an increasing realisation of the importance of sharing published data in science, which includes brain scans from neuroimaging [1]. Such sharing not only allows others to check the reproducibility of published findings, but also enables new discoveries from existing data and facilitates the creation of large imaging databases to detect subtle effects. Nevertheless, this ethical imperative for sharing can sometimes collide with the ethical imperative for patient privacy. While not normally listed as an example of “Personally Identifiable Data”, structural MRIs whose contrast is sensitive to soft tissue (e.g., T1-weighting), often include the face, which has raised concerns that research participants could be identified from a surface rendering of their images. For example, the US Health Insurance Portability and Accountability Act of 1996 (HIPAA) Privacy Rule requires that health information may not be used for research without explicit authorisation unless de-identification has been applied, including removal of “Full face photographs and any comparable images”. For this reason, T1-weighted MRIs are often “de-faced” before being shared, using one of several possible automated de-facing algorithms [2]. Although this process only affects, in theory, voxels pertaining to the face of the individual, it has been shown to affect the automatic identification and quantification of grey and white matter structures [2], probably because segmentation algorithms use the face for context. More relevant to current purposes, MRI can also be used as a secondary source of data to improve source localization in EEG and MEG and little to no attention has been paid to the effect of de-facing for this purpose.

Removing the face typically includes removing informative landmarks, such as the nose, which is a useful anchor point when coregistering a structural MRI with the location of EEG or MEG sensors via digitised “head-points”. Accurate coregistration is vital if one intends to estimate the brain sources of EEG and/or MEG data, i.e., perform source reconstruction by inverting the biophysical “forward model” that predicts how a current dipole at various positions within the brain would appear at each EEG/MEG sensor [3]. While methods exist to optimise coregistration, such as individualised head-casts (e.g., [4]), these are not always practical or appropriate for some populations, such as vulnerable patients. Thus, in most cases, coregistration is achieved by estimating the rigid-body affine mapping that minimises the error between a set of 3D points that are digitised during the MEG/EEG recording and comparable points in the MRI space. The simplest set of point consists of anatomically defined points or “fiducials”, such as the nasion and pre-auricular points, which are digitised in the MEG/EEG space and manually identified on the MRI from the same participant. However, the use of a reduced set of points increases the effect of measurement errors and, furthermore, these localisations are prone to human error. The overall coregistration error can be reduced by increasing the number of digitised points and a common practice is to digitise ~100 points distributed across the scalp and fit them to the scalp surface that can be automatically extracted from the MRI. However, since the upper part of most heads is nearly hemi-spherical, similar fits can be achieved by rotations around the centre of this hemi-sphere, i.e., “slippage” can occur owing to local minima in the error function used for coregistration. The inclusion of points on the face, particularly the nose, breaks this spherical symmetry and, therefore, helps reduce this “slippage”, i.e., improve coregistration. However, these are precisely the points that are typically unavailable in MRI space if the anatomical image has been de-faced.

Of course, one could provide the affine mapping between MEG/EEG and MRI space when sharing data by coregistering with the MRI before de-facing and sharing this affine matrix along with the de-faced MRI. However, this does not allow other users to check, or possibly improve, the coregistration with their own methods. Furthermore, de-facing can also affect the segmentation of structural MRIs (e.g., [2,5,6]), which is necessary for estimating the scalp/skull/cortical surfaces needed to construct accurate forward models. While forward models can also be shared, their construction involves many further assumptions and steps that remain contentious in the field, with different researchers preferring a range of different approaches (e.g., [7,8]). Thus, sharing a “less de-faced” image that does not affect segmentation and ideally includes the nose seems desirable. However, it remains unknown whether retaining the nose, for example, increases the identifiability of MRIs.

Many studies have asked human volunteers to match 3D-rendered MRI images without de-facing to photographs of the same individuals. For example, Prior and colleagues [9] reported that 40% of volunteers could correctly match at least 4 out of 40 individuals, while (Budin and collaborators [10]) reported that approximately 30% of individuals are at risk of recognition from 3D renderings of their MRI images. Others have used automated face-recognition algorithms and shown they can match the face that was reconstructed from an MRI FLAIR image to five different-angle photographs in 70 out of 84 participants ([11]. It is less well known how identification is impacted by various forms of de-facing (though see [10]).

A common approach to de-facing is to simply remove all voxels within a rectangular cuboid that covers the face. Though we call this “standard de-facing” here, in order to provide a reference to our new method, it is important to note that several other methods have been developed that deform or blur the face instead and it has been shown that these methods have fewer detrimental effects on subsequent segmentation of the images [10,12]. However, at the levels of deformation required to reduce identification, these methods also distort the nose; as stated by Budin and colleagues [10]: “…our method should not be used to study such topics as the shapes of noses…”.

In this work, we propose and evaluate a new method for de-identifying anatomical images while keeping specific features, i.e., the nose. To this end, we compare three versions of the same MRI images (see Figure 1A): (i) the original, unaltered MRI image, i.e., Intact; (ii) the same image, but de-identified using our proposed methodology, i.e., Trimmed; and (iii) the same image, but de-identified using a common de-facing algorithm, as provided by FreeSurfer (https://surfer.nmr.mgh.harvard.edu/fswiki/, accessed on 11 September 2022), i.e., De-faced. First, we evaluate the level of “identifiability”, asking 91 participants to match MRI images of 10 people to photographs of the same individuals. Second, we evaluate the accuracy of the coregistration of MRI images to MEG data sets, by matching digitised head points to the scalp surface extracted from the image (see Figure 1B), using over 300 MRI+MEG data pairs from the publicly available BioFIND dataset (https://www.medrxiv.org/content/10.1101/2021.05.19.21257330v1, accessed on 11 September 2022) [13]. Last, we evaluate the effect of the de-identification on the MRI tissue segmentation performed by SPM12 (https://www.fil.ion.ucl.ac.uk/spm/ (accessed on 11 September 2022)) by studying the “inner skull” surface often used for the forward model for MEG/EEG localisation.

## 2. Materials and Methods

### 2.1. MRI Data Used for the Identification Experiment

To test the effects of de-facing on manual identification, we used T1-weighted MRIs from 10 volunteers, members of the Cognition and Brain Sciences Unit (CBU) of the Medical Research Council, in Cambridge (UK). These images were accompanied by a set of 10 facial photographs (frontal view with a varying angle of 0 up to 45 degrees), one for each volunteer.

### 2.2. MEG+MRI Data Used for the Coregistration and Segmentation Experiments

The second dataset was taken from the publicly available BioFIND dataset. This dataset consists of MEG and MRI data from 324 individuals, approximately half of whom are patients with mild cognitive impairment and the rest are age-matched healthy controls. Half of the sample was acquired at the CBU and the rest at the Center for Biomedical Technology (CTB) at the Universidad Politécnica de Madrid. A full description of this dataset can be found in [13]. From the original 324 individuals in the BioFIND database, only 309 had a complete MEG-MRI pair. From these, five were discarded because of: magnetic artifacts around the mouth, likely caused by dental implants (2 individuals); having large image inhomogeneities, preventing OHBA software library (OSL) (see below) from extracting the scalp surface accurately (2 individuals); or having incorrect definition of the fiducial points in the MRI (1 individual). Therefore, the final sample consisted of 304 individuals.

### 2.3. MRI De-Facing and Trimming

We processed the anatomical images (hereby referred as Intact) in the two aforementioned datasets in the same fashion. First, we created a completely de-faced version of the image (hereby referred as De-faced), using FreeSurfer (version 5.3.0, Athinoula A. Martinos Center for Biomedical Imaging, Charlestown, MA) [14], which has shown little to no effect on some standard skull-stripping algorithms. Second, we created a “softly de-faced” (hereby referred as Trimmed) version using the proposed method, where only the soft tissue in the face is removed and the nose is kept intact. The latter trimming algorithm consists of three steps:The original MRI is nonlinearly warped into the “NY Head” tissue probability map [15], using the Unified Segmentation algorithm [16].Second, the tissue probability map is replaced with a modified version, where the face, without the nose (i.e., the area to be removed), is included as a separate tissue class. We used this map to label the different tissues in the original image.Last, the area identified as face tissue (probability higher than 10%) in the original image is removed, resulting in a trimmed MRI.

The code for these steps is available here: https://github.com/rbruna/de-facing (accessed on 11 September 2022). Figure 1A shows an Intact image (on the left), the Trimmed version (in the centre) and the De-faced version (on the right) for one of the CBU volunteers (with permission). Figure 1B shows the fiducials, head-points and scalp surface used for coregistering the MEG and MRI.

The code for all the experiments described below is available here: https://github.com/rbruna/MRI-anonymization-for-MEG-coregistration (accessed on 11 September 2022).

### 2.4. Identification Experiment

We used the first dataset to evaluate the performance of the two different de-identification approaches. For this, we generated an online experiment where a rotating (about the vertical axis) movie of one of the three rendered images (see Figure 1) of each volunteer was presented, together with a picture of each of the individuals. The task consisted of identifying which of the 10 pictures corresponded to the person shown in the movie. In order to avoid biases related to the practice, the order of presentation of the three versions (Intact, Trimmed and De-faced) was counterbalanced across participants. Last, we asked the participants to indicate their familiarity with the faces used in the experiment, since this is likely to influence the identification ability.

#### 2.4.1. Participants

The experiment was completed by two groups. One set of 50 people (23 female; mean age of 38, from the 47 who disclosed their age) were recruited from the CBU, who had varying levels of familiarity with the faces tested here. A second set of 41 people (22 female; mean age of 33, from the 40 who disclosed their age) were recruited from the CTB, who had no familiarity with the faces. Ethical approval was obtained from the Cambridge Psychology Research Ethics Committee and all participants gave their written informed consent before participation.

Data from three participants (all from the CBU group) were outliers (as defined as more than 1.5-times the interquartile range) in at least one of the conditions (one in the Trimmed condition and two in the De-faced condition). To maintain a balanced design, these three participants were removed. However, the pattern of significant results reported below remains when all the participants are included (see script “beh_data_cbu_ctb.r” in above GitHub repository).

#### 2.4.2. Methods: Experimental Design

The experiment was designed and conducted as an online study using the platform Qualtrics (https://www.qualtrics.com/, accessed on 11 September 2022). The experiment started with a familiarisation phase: participants saw 10 trials each showing a photograph of a person’s face (frontal view) and their name. Trials were displayed for 3 s during which participants were told to familiarise themselves with that face and the corresponding name.

A subsequent test phase assessed the de-facing algorithms, i.e., Intact, Trimmed and De-faced (see Figure 1), in 3 blocks, one condition per block. The CBU cohort also saw an additional 4th block which tested a Blurred condition, but as this method provided insufficient de-identification, the data are not discussed here. Block order was counterbalanced. Within a block, participants were presented with 10 trials, each showing a video of a head reconstructed from the MRI images in accord with the assessed de-facing condition. The video showed the reconstructed head spinning by 360 degrees about a vertical access for 14.4 s. Appendix A shows stills of the experiment shown to the volunteers. After the completion of 2 full rotations, the head disappeared. Participants were then asked to identify the person presented in the scan as best as they can, using an alternative forced choice test which displayed all 10 pre-familiarised faces-name options. These options remained on the screen until a response was given and once a choice was logged via mouse click on target face-name pair, the next trial commenced.

After the test phase, participants once more saw the photos of all 10 faces, one at a time and were asked to rate their pre-experimental familiarity with each face using a scale ranging from zero (not at all familiar) to ten (extremely familiar). The experiment then concluded with a short survey collecting demographic information and feedback asking whether participants used any strategies or encountered technical difficulties.

#### 2.4.3. Statistical Analysis

Statistical analysis was performed in R. The main comparison was based on a mixed-effects ANOVA, using the Greenhouse–Geisser correction (epsilon reported below) to correct for possible nonsphericity. Follow-on, pairwise comparisons were based on paired T-tests evaluated with two tails, corrected for unequal variance using the Welch’s method. Last, the evidence in favour of the null hypothesis was tested using Bayes factors (BF01), by means of the “ttestBF” function in the “BayesFactor” package. All these comparisons can be replicated using the GitHub script “beh_data_cbu_ctb.r”.

### 2.5. MEG-MRI Coregistration Experiment

We coregistered the MRI and MEG head shape using a two-step procedure, as recommended by the OSL software (https://ohba-analysis.github.io/osl-docs/matlab/osl_example_coregistration.html, accessed on 11 September 2022). In a first step, the two images (MRI and MEG head shape) were coregistered using a linear transformation between the fiducial points (nasion and both pre-auricular points) in both images. Then, we used OSL’s RHINO (i.e., Registration using Head shapes Including Nose) function to obtain the best fit between the head shape points and the scalp surface extracted from the MRI (see Figure 1 for an example). For this, RHINO uses an iterative closest points (ICPs) approach across 10 random initialisations.

We evaluated the coregistration error as the mean Euclidean distance, after RHINO, between the fiducials defined in the MEG head shape and those defined in the MRI. As distance is always positive and, therefore, the distance distributions are positively skewed, we compared the results for the different de-identification approaches using a two-tailed Wilcoxon signed-rank test.

### 2.6. MRI Segmentation Experiment

Source reconstruction is an inverse solution for the origin of the electrophysiological signal recorded outside the head and as such requires a forward model describing the propagation of this signal from the brain sources [17]. This forward model depends on the structure and tissues of the head and is usually based on a segmentation of an anatomical image, typically an MRI. In the case of MEG, the most important structure is the inner skull and many forward models consider a single-shell head model, where the inner skull serves as interface between a homogeneous conducting medium (inside) and a nonconducting medium (outside). Then, the forward model is created by estimating the lead fields, i.e., the signal that would be measured at each sensor from an electrical dipole at each possible source location [18].

The automatic extraction of the inner skull surface is more difficult than the extraction of the scalp surface, as several tissues with similar T1-weighted MRI signal level MRI are involved in this interface. Thus, rather than extracting this surface directly (as RHINO does to extract the scalp surface above), SPM approximates the inner skull by taking an existing surface from a template and warping it to closely match the subject-specific anatomy [19]. This warping is based on the nonlinear matching between the MRI and a tissue probability map and this matching can be affected by the de-facing process.

To evaluate whether this is the case and quantify the differences between the surfaces generated for each image, we calculated the mean error between the 2562 vertices defining this surface. Here, we quantified this error for the Trimmed and De-faced images as the mean distance between the vertices defined for these images and those defined from the original one image. As a change in the position (a translation) of the surface will likely have less impact in the head model than a change in its shape, we also applied a rigid-body ICP coregistration algorithm to the inner skull surfaces, such that any residual difference is due to different shapes of the surfaces themselves (i.e., a different warping between the template and the image). As the forward model is highly dependent on this surface, any difference in its shape will necessarily entail differences in the forward model and, therefore, affect the reconstruction of the sources generating the sensor data, such as for the coregistration error above, the mean error for the different de-identification approaches was compared using a two-tailed Wilcoxon signed-rank test.

## 3. Results

This section may be divided by subheadings. It should provide a concise and precise description of the experimental results, their interpretation, as well as the experimental conclusions that can be drawn.

### 3.1. Identification Accuracy

As expected, mean identification accuracy was highest for Intact images and worst for De-faced images (Figure 2A), with that for Trimmed images falling in between. Mean accuracy was also higher for the CBU group than the CTB group, also as expected, given that the CBU group had personal familiarity with some of the people whose images were used. Indeed, individual ratings of familiarity, provided by the CBU participants, were positively related to accuracy at the single-trial level, as indicated in the Appendix A. An initial Analysis of Variance (ANOVA) confirmed significant main effects of condition (*F*(2,172, epsilon = 0.87) = 46.9, *p* < 0.001) and group (*F*(1,86) = 16.0, *p* < 0.001). The group-by-condition interaction approached significance (*F*(2,172, epsilon = 0.87) = 2.47, *p* = 0.094), though this was most likely driven by floor-effects, as the accuracy in the De-faced condition approached chance for both groups (Figure 2A).

However, the above analysis disguises an effect of block order on accuracy. When dividing the data according to an (between-participant) “order” factor, defined by whether or not a condition was the first one attempted, performance improved after first exposure to a condition, at least for the Intact and Trimmed faces (Figure 2B; performance on the De-faced condition remained close to chance). This suggests that the most appropriate test of identification is to examine first blocks only, so that the accuracy of, for example, Trimmed faces is not contaminated by having previously seen the Intact versions.

When analysing first blocks only (blue line in Figure 2B), pairwise T-tests showed a reliable difference between Intact and Trimmed images (*t*(49.0) = 2.34, *p* = 0.023), but not between Trimmed and De-faced images (*t*(50.6) = 0.056, *p* = 0.956). A Bayesian T-test provided a Bayes Factor (BF01) of 3.61, i.e., moderate evidence [20] for no difference in identification accuracy for (first exposure to) Trimmed and De-faced images.

Finally, note that performance in all first-only conditions was significantly greater than a chance level of 0.1, even for De-faced images (all *t*s > 4.62, all *p*s < 0.001). This may reflect the use of distinctive features (e.g., the shape of the ears or the head size) that remain, for a subset of the images, even after de-facing (see Appendix A). For example, if those features were sufficient to identify the sex (but not the identity) of the image, then chance would be 0.2 instead of 0.1. While Intact images were above this chance level of 0.2 (*t*(26) = 4.11, *p* < 0.001), Trimmed and De-faced images were not (*t*s < 1.54, *p*s > 0.13), although there was insufficient evidence to claim performance came from this chance level (all BF01’s < 1.96).

### 3.2. MRI-MEG Coregistration

We summarised coregistration error as the mean Euclidean distance, after coregistration, between the three anatomically defined fiducials (nasion plus left and right pre-auricular points) defined in the MRI and their homologues in the MEG head shape. These points are indicated by a researcher when examining the MRI and when digitising the head for MEG, respectively. Given that there is nearly always manual error in estimating these points, we calculated a baseline error by coregistering the three digitised MEG fiducials directly to the three corresponding points on the MRI. The distribution of errors over *n* = 304 participants is shown in the left distribution in Figure 3A, which ranged from 0.66 mm to 10.97 mm, with a median error of 4.24 mm.

We then calculated the same fiducial registration error after coregistering all the digitised head points (excluding the three fiducials) to the scalp surface extracted from each of the three anatomical images (next three distributions in Figure 3A). Not surprisingly, the fiducial error increased for all conditions relative to directly coregistering the fiducials, but note that this does not mean that coregistration with head points is always worse; the manual error in marking fiducials means that the true coregistration is unknown and the error from directly coregistering the fiducials just provides a lower limit. The error differed little between the Intact (median = 14.56 mm) and Trimmed images (median = 14.31 mm), though, it was, in fact, significantly lower (*p* = 0.017) for the Trimmed images. This difference would not survive correction for the multiple pairwise comparisons performed here, so may be a chance finding. Nonetheless, the results demonstrate that our Trimming method does not harm coregistration, at least when digitised face points are placed around the nose. Importantly, the coregistration error was significantly greater in the De-faced (median = 21.29 mm) than either Intact or Trimmed conditions (*p* < 0.001), demonstrating that de-faced MRIs should not be coregistered to a set of digitised points that include the face. Nonetheless, this is not surprising, because the presence of digitised points on the nose (nose-points) will tend to “drag” the MEG data posteriorly relative to the De-faced MRI data, so as to minimise the error between those nose points and the most anterior part of the (not anatomically correct) scalp surface in the absence of the face.

Figure 3B, therefore, shows the same error after excluding these nose points (by removing digitised points with y > 0 and z < 0 in the MEG space, i.e., relative to the origin defined by the nasion and pre-articular points). As expected, the error is substantially reduced for the De-faced condition (median = 15.30 mm). Conversely, the error for the Intact (median = 15.62 mm) and Trimmed (median = 15.44 mm) images is slightly increased, because the remaining face surface (particularly the nose) can potentially bias coregistration by “pulling” the remaining head points anteriorly (or, at least, increase chances of local minima). Interestingly, the error is significantly lower in both the Trimmed (*p* = 0.0016) and De-faced (*p* = 0.0015) images than in the Intact one.

Most important of all, however, is to compare coregistration when the nose points are included and the nose is present (in the Trimmed MRI) with that when the nose points are excluded and the nose is absent (from the De-faced MRI)—i.e., the third and eighth distributions in Figure 3. Here, the median error was significantly (*p* < 0.001) smaller in the former case (median Trimmed error with nose = 14.31 mm, median De-faced error without nose = 15.30 mm). This suggests that the presence of the nose can improve coregistration, most likely functioning as an anchor to prevent slippage into local minima.

### 3.3. MRI Segmentation

Removing the face can also affect automated MRI segmentation techniques, which are often used to construct the surfaces used for MEG forward modelling, such as the inner skull surface (see Section 2). In SPM, a “canonical” inner skull surface is created for each participant by warping an inner skull surface from a template (MNI) brain into the native MRI space, through inverting the normalisation parameters (warps) determined by matching their MRI to that template MRI. Removing the face can affect estimation of those normalisation parameters and, hence, affect the forward model used for MEG, regardless of whether the abovementioned MRI-MEG coregistration is also affected. We examined this possibility by estimating the mean distance between the 2562 vertices of the canonical inner skull surface for each participant created from their Intact MRI (considered as gold standard) versus their Trimmed and De-faced MRIs.

As can be seen in Figure 4, the shape of the inner skull surfaces does differ between conditions. Moreover, as shown in the left panel of Figure 4B, the difference between the surfaces extracted from the Intact and De-faced images (median = 0.49 mm) is significantly larger (*p* < 0.001) than the difference between those extracted from the Intact and Trimmed images (median = 0.20 mm).

The images in Figure 4A show that, as expected, the difference in the head models is more prominent in frontal areas, closest to the modified parts of the images, than in the rest of the head. The overall error, therefore, can be reduced if the surface is slightly displaced posteriorly. As this translation will likely have less effect on the forward model than the shape difference, we repeated the statistical analysis after rigid-body coregistration of the surfaces coming from both the Trimmed and De-faced images to the original surface. As shown in right panel of Figure 4B, the errors were reduced, but the Trimmed images still introduced significantly (*p* < 0.001) less difference from the Intact images (0.19 mm) than the De-faced images did (0.39 mm).

## 4. Discussion

MEG (and EEG) data are acquired using sensors placed outside the head and used to infer what is happening inside of the head, i.e., in the brain. This often requires the task of source reconstruction, an inverse problem that requires an accurate forward model to work well. This forward model can be constructed from a template head, but its accuracy improves when using an individual’s anatomy [21]. This anatomy normally comes from an MRI image that is acquired separately from the MEG/EEG data and the accuracy of the forward models depends strongly on precise coregistration of the two types of data [3]. For this, the nose plays an important role, as it serves as a landmark that breaks the otherwise near-spherical symmetry of the head. Nevertheless, the face, in general, and the nose, in particular, are usually disregarded when analysing MRI data on their own. Therefore, when MRI data are shared, following the open science philosophy [1], the face is usually removed (“de-faced”) to help protect the privacy of participants [22].

In this work, we studied the role of the nose in the MEG-MRI coregistration, measuring the coregistration error for (i) non-de-faced (Intact) images, (ii) images after a standard anonymization algorithm (De-faced) and (iii) images after a new MEG-friendly de-facing algorithm (Trimmed), where the nose is kept, but other facial features are removed. Our results show that, when using the less aggressive algorithm, the Trimmed images can be coregistered to the MEG data with less error than the De-faced images. This improvement in coregistration is likely due to the inclusion of the nose in the image, because the difference between approaches disappears when the points corresponding to the nose and the face are removed from the MEG head points. Nevertheless, when we compared the error with or without considering the nose, the former is significantly lower, indicating that the presence of the nose does contribute to better coregistration. These results support the idea than the nose is an important landmark for a correct coregistration of the images and, from this perspective, it should be included in open datasets, both in the MRI and in the digitised MEG head points.

Nevertheless, the cost of improved coregistration by Trimming rather than De-facing could be a less effective de-identification of MRI images. To check whether this is the case, we asked 91 participants to identify 10 volunteers, pairing pictures of them with 3D renderings of their MRI images. When controlling for practice effects by considering only the first time an MRI image was seen (so that a Trimmed image, for example, did not benefit from having previously seen the Intact version), the results showed that Trimming provided a level of de-identification similar to that provided by the conventional De-facing. Indeed, both Trimmed and De-faced images showed levels of de-identification significantly lower than Intact images, with no difference between them (BF01 = 3.61). When including images that had been seen before in other conditions, the accuracy of the identification was increased for Trimmed relative to De-faced images, but in a real-life situation, the observer would have access to a single version of the image, namely the Trimmed or De-faced one, and, therefore, identification would not benefit from having seen an Intact version too.

Last, we evaluated the effect of the de-identification in the tissue segmentation, paying attention to the shape of the inner skull surface generated from the de-identified images. When using a common approach in the SPM software package of deforming a template inner skull surface, we found that the shape of this surface was significantly more affected when using De-faced images than when using Trimmed images (relative to the Intact image). However, even though the difference is smaller, the surface extracted from the Trimmed image was not identical to that extracted from the Intact one, suggesting that some information is altered after both types of de-identification.

Altogether, our results indicate that our proposed method is less aggressive, keeping important information that is removed or modified when using a standard de-facing, but still maintains a similar level of de-identification to the standard approach. Nevertheless, some points merit discussion, both regarding our particular results and de-identification of anatomical images in general.

### 4.1. MEG-MRI Coregistration

Our main experiment was based on the idea that the nose can serve as a good reference when coregistering MEG data to MRI anatomical images. From experience, this seems to be the case: head points for MEG are taken primarily from the top of the head and this part of the anatomy is quite spherical in shape, with large rotational symmetry. Therefore, trying to fit these points to the scalp extracted from an MRI is quite difficult without some “boundary conditions”. As the fiducials are fixed points in the anatomy, they serve as these boundary conditions: the preauricular points define an axis (the *x*-axis, in Neuromag coordinates) and the nasion defines the angle of rotation. Three points are enough to determine a transformation matrix in 3D and a correct definition of these fiducials should be enough for an accurate coregistration. Nevertheless, the definition of these points is prone to error: measurement errors at the time of MEG acquisition and/or errors in identifying the fiducials on an MRI (e.g., due to subjective differences between operators).

Note that incorrect positioning of these fiducial points can produce large errors. For example, even if the nasion is correctly placed but both preauricular points have a consistent error of 5 mm superior to their true position, the error on the back of the head would be double that (i.e., head points would be 10 mm superior to their real position), given that the preauricular points define the centre of the head (A fourth anatomical landmark of the “inion”, at the back of the head, is sometimes used, which should provide greater robustness, though the precise location of the inion is not easy to identify on all individuals). This error can be identified and compensated by using more head points in the coregistration and this is the basis of OSL’s RHINO algorithm used here. However, it is difficult to establish a ground truth for “perfect” coregistration. In our case, we quantified the error as the difference between the fiducials in the MRI and the MEG, which are, as indicated above, prone to error. Even so, it should be remembered that this “more accurate” coregistration is only a relative measure and may include other sources of error common to all types of MRI/algorithms.

### 4.2. De-Identification

The level of de-identification achieved by the Trimmed images was similar to that achieved by the standard De-faced ones. Nevertheless, it is important to note that overall accuracy in our identification experiment was relatively high, even for the De-faced images. If the participants guessed the identity randomly, we would expect a chance accuracy of 10% and the accuracy of our participants was significantly higher than that. However, accuracy levels must be interpreted in this context of a limited set of 10 identities and cannot be extrapolated directly to other situations in which identification is a concern. This is because there are likely to be some features (e.g., the shape of the ears) that are unique to 1 of these 10 identities, but that would not be so distinctive if an identity could come from a much larger set (e.g., all individuals one knows). Indeed, some of the present images were more often identified than others, likely because of such distinctive features (see Appendix A). Another reason for the relatively high performance in the present experiment might be that there were aspects of the images (e.g., overall head size) that were strongly associated with males versus females and extracting only this sex information would be sufficient to halve the set of possible faces to match, i.e., increase the chance level to 20%, even if identification of a unique individual never occurred. In other words, the relatively high identification level in our experiment is unlikely to apply to more real-world situations where an MRI could come from a much larger and heterogeneous set of individuals.

### 4.3. MRI Segmentation

In our experiment, the estimation of the inner skull surface from the Trimmed image was closer to the original one than that estimated from the De-faced image. However, it was not identical, indicating that although smaller than the conventional approach, the suggested method might affect the segmentation of the MRI image. Future work should test how much effect this type of error has on inverse solutions. It is important to note that we used a common approximation to calculate the inner skull surface, as implemented in the SPM software, which is based on warping (inverse normalisation) of a template mesh. Other MRI packages, such as FreeSurfer, attempt to extract the inner skull surface directly from the MRI. However, this can be prone to error and other studies have shown that de-facing can affect a range of MRI segmentation algorithms [2,5,6]. Indeed, even different images from the same participant can give rise to slightly different segmentation outputs [23,24]. As we only have one scan per participant, it is not possible for us to determine if the difference between the inner skull surfaces extracted from the Intact and Trimmed images falls within this margin of error. If possible, future efforts should address this issue, in order to determine a “healthy” limit for the variability in the definition of the surfaces.

## 5. Conclusions

Our results show that de-identification, either removing or keeping the nose, significantly hinders the ability of human participants to pair photographs of volunteers to their MRIs. However, it is possible that automated pattern-matching (machine-learning) algorithms could do better than our humans in matching features extracted from an MRI (either de-faced or not) to those extracted from a 2D photograph (e.g., Schwarz et al., 2019). Indeed, it is possible that just the precise shape of the nose (as present in our Trimmed images) is sufficient to match the rendered MRI images, e.g., to a series of facial photographs from multiple angles, at a high level of accuracy. Thus, it is important to note that complete de-identification of structural MRI images is unlikely to be possible. Indeed, even without the face, it seems likely that an individual could be identified via the precise shape of their brain, such as their cortical (sulcal/gyral) folding (or other “cortical fingerprints” [25,26]). While this requires already having that information (e.g., from another anatomical image), this matching of brains would, in principle, allow someone to recover other sensitive information also associated with a given research participant when sharing data (e.g., their IQ scores). A recent study suggesting sharing a warped version of the MRI, which would be difficult to identify in terms of cortical fingerprinting, was shown to have little effect on MEG source reconstruction, since the warping maintained sufficient geometrical similarity to the original MRI [27]. However, because this warping affected structural analysis of the MRI, the authors advised that sharing of the original MRI is preferable when consent is available. Therefore, rather than aiming for complete de-identification of brain images, which might never be possible or feasible, it is important to explain to the participants the risk of identification before they consent to their data being shared [22]. If that informed consent is given, then there are normally no ethical or legal concerns about sharing research data. Thus, any method, such as de-facing, which can reduce the chances of identification (even if not eliminate that risk), may help reassure participants sufficiently that they are prepared to provide consent for data sharing. In other words, we expect that participants will be more likely to agree to data sharing if they are informed that most of their face will be removed from the images (even if not the nose), such that identification is impaired, even if not fully prevented, as demonstrated by the results of the identification experiment described here.

## Figures and Tables

**Figure 1 bioengineering-09-00591-f001:**
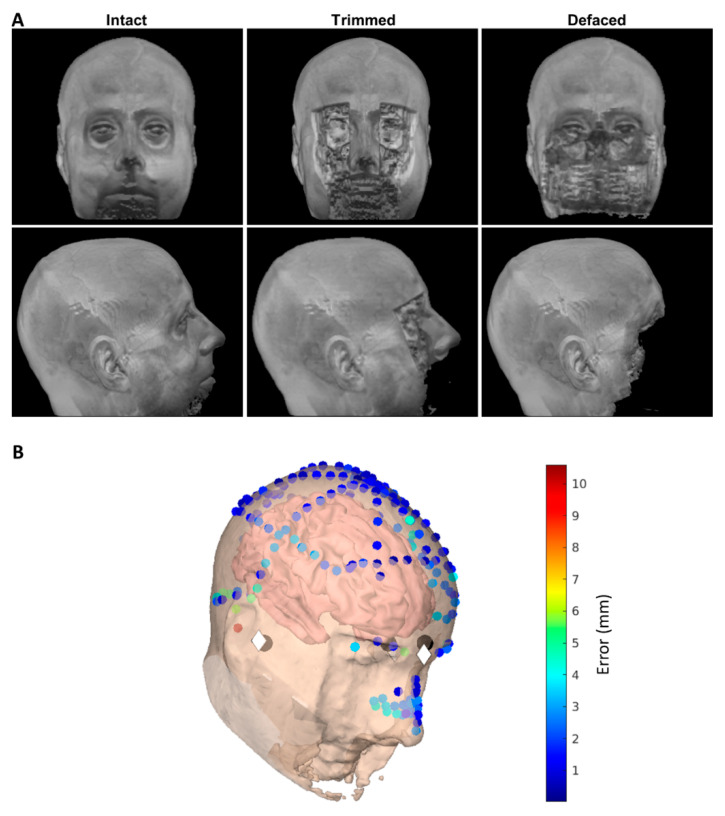
(**A**) Example surface renderings of original, “Trimmed” (that maintains the nose) and standard “De-faced” MRI (columns) from front (top row) or side (bottom row) views. (**B**) locations on semi-transparent scalp surface from trimmed MRI in (Panel (**A**)) (plus cortex inside) of three anatomical fiducials digitized during an MEG experiment (dark grey circles), same fiducials indicated on MRI (white diamonds) and head- and nose-points digitized during MEG (small, coloured circles). Colour of head/nose-point circles shows error (in mm) from scalp surface after coregistration.

**Figure 2 bioengineering-09-00591-f002:**
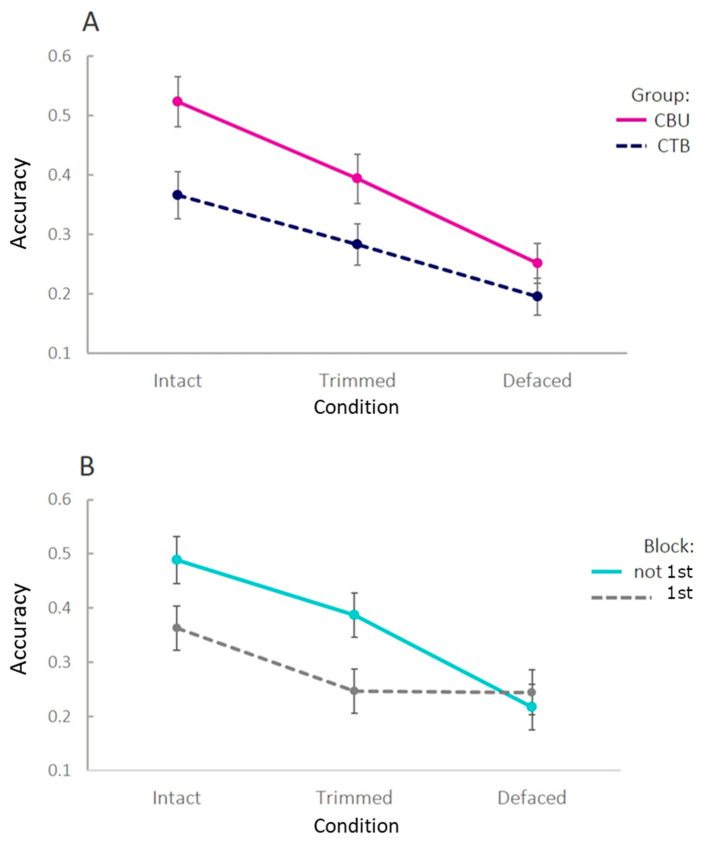
Identification accuracy as function of Condition and Participant Group (Panel (**A**)) or whether the condition was run as the first block or not (Panel (**B**)).

**Figure 3 bioengineering-09-00591-f003:**
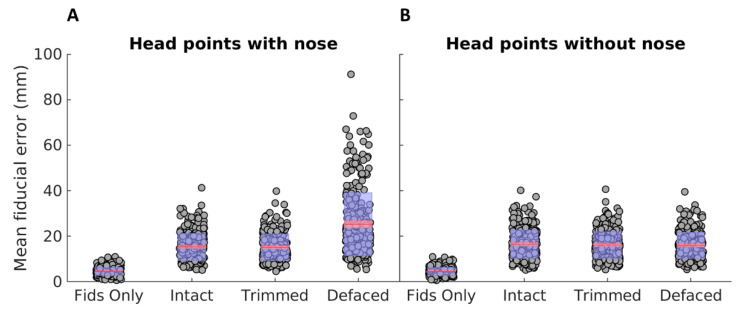
(**A**) Mean difference in position (error) between the fiducials defined in MRI and the fiducials defined in MEG, when both images were coregistered using OSL’s RHINO function. (**B**) Same, but after removing the nose points from the MEG head points. The figure shows the median error (in red, SEM as a red shadowed area) and the standard deviation (blue shadowed area) for the coregistration using only the fiducials (Fids Only) or after coregistering the MEG head points to the “Intact”, “Trimmed” or “De-faced” images.

**Figure 4 bioengineering-09-00591-f004:**
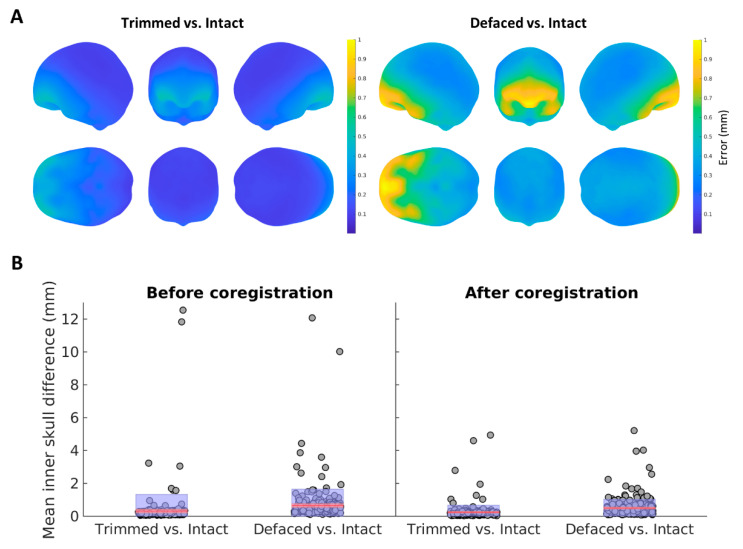
(**A**) Distribution of the error (in millimetres) between the inner skull surface generated from one participant’s Intact MRI and that generated from either their Trimmed (**left**) or De-faced (**right**) MRIs. (**B**) Distribution of mean errors across participants before (**left**) and after (**right**) applying a rigid body coregistration of the meshes. The figure shows the median error (in red) and the standard deviation (blue shadowed area) for the mean difference between the surface mesh generated from the Intact MRI and either the Trimmed or the De-faced MRIs.

## Data Availability

The data used in this work ae publicly available (under request) under the terms of the BioFIND project. See https://www.medrxiv.org/content/10.1101/2021.05.19.21257330v1 for details, accessed on 11 September 2022. The code used in this work is publicly available in https://github.com/rbruna/MRI-anonymization-for-MEG-coregistration, accessed on 11 September 2022.

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
