# Peer review of "Modified MRI Anonymization (De-Facing) for Improved MEG Coregistration"

_bioengineering, 2022, doi:10.3390/bioengineering9100591_

Round 1

Reviewer 1 Report

Title : Modified MRI anonymization (defacing) for improved MEG coregistration

This is a well written paper that details the role of the nose for creating a better coregistration between the MEG and the MRI.  The statistics are clear.

Specific issues

Page 2

 Line 47 insert the word ‘such’ before ‘as the nose’

Line 63 delete the word ‘further’

Line 68 should  read ‘,and helps to reduce “slippage”..’

Line 95 insert the word ‘have’ before ‘shown’.

Page 3

Line 102 In text the figure indicates ‘I’, ‘ii’, ‘iii’, but these not indicated in the figure.  Instead of i replace with ‘intact’, ii replace with trim and iii with defaced as headings above the MRI images.  Also the scale  needs a label to indicate what the colors represent.  The Grey Diamonds are very hard to see.

Page 5

Line 168 replace the word ‘showed’ with ‘shown’.

Page 7

Line 283 it is unclear what the meaning of “defaced condition remained close to floor”

Page 9

Figure 3 the figure is missing the A and the B label.

page 10

Figure 4 A  should have a heading ‘trimmed’ and ‘defaced’ above the blue images.

Page 11

Line 398 insert the word ‘the’ in front of ‘MRI image’.

Author Response

Reviewer #1

This is a well written paper that details the role of the nose for creating a better coregistration between the MEG and the MRI.  The statistics are clear.

We are thankful to the reviewer for her/his kind words. We will try to address his/her comments below, in blue ink. If we make any change in the text of the manuscript, we will also mark it, but in red ink, to make it easily identifiable.

Specific issues
Page 2
Line 47 insert the word ‘such’ before ‘as the nose’

We have added the word such and, indeed, the sentence reads much better.

Line 63 delete the word ‘further’

We have removed, we thank the reviewer for pointing it out.

Line 68 should read ‘,and helps to reduce “slippage”..’

We have replaced “so” by “therefore”, as we indeed consider the use of this linker helps the reader to understand the sentence. Of course, we could be wrong. We have, however, replaced added the “s” to “helps”, and apologize for the typo.

Line 95 insert the word ‘have’ before ‘shown’.

We agree with the reviewer, the sentence was not correctly written. Combining her/his suggestion with the on provided by Reviewer #2 we have replaced it by “it has been shown”.

Page 3
Line 102 In text the figure indicates ‘I’, ‘ii’, ‘iii’, but these not indicated in the figure.  Instead of i replace with ‘intact’, ii replace with trim and iii with defaced as headings above the MRI images.  Also the scale needs a label to indicate what the colors represent.  The Grey Diamonds are very hard to see.

We apologize for this omission, we have added the labels to panel A, and we have added this information to the text. However, we kept the numeration as Roman numerals, and added the label of the images as follows:

To this end, we compare three versions of the same MRI images (see Figure 1A): i) the original, unaltered MRI image, i.e., Intact; ii) the same image, but de-identified using our proposed methodology, i.e., Trimmed; and iii) the same image, but de-identified using a common defacing algorithm, as provided by FreeSurfer (https://surfer.nmr.mgh.harvard.edu/fswiki/), i.e., Defaced.

Regarding the diamonds, we have replaced the ones in the “camera” side of the head by flat white diamonds, to make them more visible.

Page 5
Line 168 replace the word ‘showed’ with ‘shown’.

We apologize for this error. We have fixed it in the manuscript.

Page 7
Line 283 it is unclear what the meaning of “defaced condition remained close to floor”

We see now that we were not clear, and we apologize. We were referring to the theoretical “chance” level as “floor”. To be clearer, we have replaced “floor” by “chance” in the sentence.

Page 9
Figure 3 the figure is missing the A and the B label.

We apologize for this omission. We have now added the A and B labels.

page 10
Figure 4 A should have a heading ‘trimmed’ and ‘defaced’ above the blue images.

We apologize for the omission. We have now added the label to the figure.

Page 11
Line 398 insert the word ‘the’ in front of ‘MRI image’.

We have included the word “an” instead of “the”, as we are not referring to a specific image. But we thank the reviewer for pointing this out.

Reviewer 2 Report

In this work, the authors address the impact of MRI anonymisation on MEG coregistration. Anonymisation is crucial in the sharing of data for collaboration and Open Science. However, the removal of key features (e.g., the nose) can lead to the head shape being overly symmetric, leading to an incorrect coregistration. The authors propose a solution that removes identifiable facial features but leaves the nose intact, thus reducing the symmetry of the head shape and improving the coregistration. The authors also explore the effect of MRI defacing on segmentation accuracy, as well as the effectiveness of the anonymisation while leaving a key facial feature in place. 

The work presented here is thorough and will be of great benefit in the sharing of structural MRI data. The insights offered on segmentation errors was particularly useful for me, as these explain many issues I have seen in my own work. There is certainly an argument that if you know someone well enough, you will still be able to recognise them, but the results shown from the Identification Experiment are indeed very promising from a data protection viewpoint. 

I have no major corrections, only a small selection of edits/suggestions.

1) Page 2, Line 95: I would suggest adding "and it has been shown that these methods" for ease of reading.

2) Page 4, Line 138: I think the OSL acronym should be defined

3) Page 4, Line 152: I think noise should be nose?

4) Page 5, 2.4.2 Methods: Experimental Design: It could be useful for some stills of the online study to be shown in a figure to further clarify the experimental design.

5) Page 5, Line 196: "...de-identification, and the data..." The "and" is not needed here.

6) Page 5, Line 206: "...familiarity with each faces using..." Singular face. 

7) Page 6, Line 223: The RHINO acronym should be defined as there is another 3D software package by the same name.

8) Page 6, Line 226: Maybe claify what the random initialisations are? (i.e., are they rotational transformations, translational, or both? To what extent in degrees and mm?)

9) Page 8, Figure 2: Panel A x-axis should be labelled "Condition". Axis labels should probably be capitalised too.

10) Page 8, Line 305: For the mean Euclidean distance, is this the most appropriate measure? If the LPA and RPA points are spot on, but the nasion points disagree, won't the mean be thrown off? A justification on this might be useful.

11) Page 10, Figure 4: Panel A needs labels, e.g., Trimmed and Defaced above each group, as well as a label and units for the colour bar. 

Author Response

Reviewer #2

In this work, the authors address the impact of MRI anonymisation on MEG coregistration. Anonymisation is crucial in the sharing of data for collaboration and Open Science. However, the removal of key features (e.g., the nose) can lead to the head shape being overly symmetric, leading to an incorrect coregistration. The authors propose a solution that removes identifiable facial features but leaves the nose intact, thus reducing the symmetry of the head shape and improving the coregistration. The authors also explore the effect of MRI defacing on segmentation accuracy, as well as the effectiveness of the anonymisation while leaving a key facial feature in place.

The work presented here is thorough and will be of great benefit in the sharing of structural MRI data. The insights offered on segmentation errors was particularly useful for me, as these explain many issues I have seen in my own work. There is certainly an argument that if you know someone well enough, you will still be able to recognise them, but the results shown from the Identification Experiment are indeed very promising from a data protection viewpoint.

We are thankful to the reviewer for his/her kind words, and we are happy she/he shares our interest for both Open Science and privacy. We will try to address his/hers comments below, in blue ink. If we include any except of the text it will be written in black ink with an indentation, and any new text, either in the excerpt or in the manuscript, will be marked in red ink.

I have no major corrections, only a small selection of edits/suggestions.
1) Page 2, Line 95: I would suggest adding "and it has been shown that these methods" for ease of reading.

We have modified the sentence as the reviewer recommends and, indeed, the sentence reads much better now. We are thankful to him/her for her/his suggestion.

2) Page 4, Line 138: I think the OSL acronym should be defined

It is indeed unfortunate that we mention OSL before introducing it. We have replaced “OSL” by “OHBA software library (OSL)”, although the software is “formally” introduced in the Methods section.

3) Page 4, Line 152: I think noise should be nose?

The reviewer is right, we have corrected the typo.

4) Page 5, 2.4.2 Methods: Experimental Design: It could be useful for some stills of the online study to be shown in a figure to further clarify the experimental design.

We agree with re reviewer that more information about the online experiment is interesting, so we provide in the Supplementary materials (Figure S2) still images for two of the people who nicely provided their image for the experiment. Unfortunately, we only have permission from these two individuals to show their pictures and MRIs outside of the experiment, but we think these two examples are illustrative of the experiment.

5) Page 5, Line 196: "...de-identification, and the data..." The "and" is not needed here.

The reviewer is completely right, we apologize.

6) Page 5, Line 206: "...familiarity with each faces using..." Singular face.

We apologize for this typo. It is now corrected.

7) Page 6, Line 223: The RHINO acronym should be defined as there is another 3D software package by the same name.

We want to thank the reviewer for pointing this out, as we did not know of this 3D package. We have added the definition of the acronym to avoid misinterpretations.

8) Page 6, Line 226: Maybe claify what the random initialisations are? (i.e., are they rotational transformations, translational, or both? To what extent in degrees and mm?)

The number of random initialisations is an internal parameter of the RHINO software, and it is not described in detail. Looking at the code, it seems to apply a random rotation and translation to the result (up to 15 degrees rotations and up to 10 mm translations) and pass it again through the algorithm.

The use of 10 random initialisations is the default in RHINO, and is the recommended in their guides.

9) Page 8, Figure 2: Panel A x-axis should be labelled "Condition". Axis labels should probably be capitalised too.

We thank the reviewer for pointing this out. We have modified the labels according to her/his suggestions. We omitted the label in the X-axis of the A panel to make the image clearer, as the horizontal axis is the same for both panels. But we see now that, being the panels treated independent as A and B, this omission is more confusing than helping.

10) Page 8, Line 305: For the mean Euclidean distance, is this the most appropriate measure? If the LPA and RPA points are spot on, but the nasion points disagree, won't the mean be thrown off? A justification on this might be useful.

The Euclidean distance seems the most natural measure of error in this situation (equivalent to the root-mean-square error often used in optimisation). If the reviewer is suggesting that another summary statistic like median might be more appropriate than the mean when combining the 3 fiducials (e.g., for example given), then this is indeed possible, but we suspect that examples like that given when one fiducial fit is much worse than the other two is quite rare, so the mean and median will often be aligned. We note also that the same error (cost) function, i.e., the root-mean-squared of the Euclidean distance, is used by RHINO to fit to the head shape to the volumetric image.

The automatic evaluation of the quality of a MEG to MRI realignment is not easy; if one metric of real error was available, the optimal alignment could be achieved by minimizing this error. Here, we assume that the anatomical landmarks are correctly placed (although that is not exactly the case, as the error with fiducials only is not zero), and therefore, a displacement in the MEG landmarks, compared to the MRI ones, must be the result of an error in realignment.

11) Page 10, Figure 4: Panel A needs labels, e.g., Trimmed and Defaced above each group, as well as a label and units for the colour bar.

We apologize for the omission. We have now added the label and units to the figure.

Reviewer 3 Report

In this paper, the authors discussed the trade-off between de-identification of the MRI data in terms of privacy protection of participants and the accuracy of the MEG/EEG source reconstruction, and proposed a new method to trim the MRI data consdering both of them. The perspective of the authors is unique, practical and worthy of publication.

There were two tiny points of concern. If they are typos, please modify them before it's published.
1. On L. 152, is 'without the noise' typo for 'without the nose"?
2. On L. 390, why 'the' of 'the surface' is in bold?

Author Response

Reviewer #3

In this paper, the authors discussed the trade-off between de-identification of the MRI data in terms of privacy protection of participants and the accuracy of the MEG/EEG source reconstruction, and proposed a new method to trim the MRI data consdering both of them. The perspective of the authors is unique, practical and worthy of publication.

We are thankful to the reviewer for his/her kind words, and we are happy she/he shares our interest in the topic of our work. We will address his/her comments here in blue ink, and the changes in the manuscript will be marked in red ink.

There were two tiny points of concern. If they are typos, please modify them before it's published.
1. On L. 152, is 'without the noise' typo for 'without the nose"?
2. On L. 390, why 'the' of 'the surface' is in bold?

Those were indeed typos, and we are thankful to the reviewer for pointing them out. We have corrected them in the current version of the manuscript.